# Quantitative Examination of the Inclusion and the Rotated Bending Fatigue Behavior of SAE52100

Xueliang An [1,2], Zhiyue Shi [2,3], Haifeng Xu [2], Cunyu Wang [2], Yuhui Wang [1], Wenquan Cao [2,*] and Jinku Yu [1,*]

1   State Key Laboratory of Metastable Materials Science and Technology, Yanshan University, Qinhuangdao 066004, China; anxueliang@stumail.ysu.edu.cn (X.A.); yhwang@ysu.edu.cn (Y.W.)
2   Central Iron and Steel Research Institute (CISRI), Beijing 100081, China; b20190517@xs.ustb.edu.cn (Z.S.); xuhaifeng228@163.com (H.X.); wang_cunyu@126.com (C.W.)
3   School of Materials Science and Engineering, University of Science and Technology of Beijing, Beijing 100085, China
*   Correspondence: caowenquan@nercast.com (W.C.); yujinku@ysu.edu.cn (J.Y.); Tel.: +86-106-2182-628 (W.C.); +86-0335-807-4792 (J.Y.)

**Abstract:** This study investigated the effect of maximum inclusion on the life of SAE52100 bearing steel processed by two different melting routes, vacuum induction melting plus electroslag remelting (VIM + ESR), and basic oxygen furnace plus ladle furnace plus vacuum degassing process (BOF + LF + RH) by the metallographic method, Aspex explorer, and rotated bending fatigue test. The rotated bending method was applied to examine the maximum inclusion size in a satisfactory manner, whereas both the metallographic method and Aspex explorer underestimated the result. Regardless of the characterization methods, the results show that the total number of inclusions in VIM + ESR melted steel is significantly higher than that in BOF + LF + RH processed steel, but the maximum inclusion size of VIM + ESR melted steel is significantly smaller than that of the BOF + LF + RH degassed steel. The distribution of the maximum inclusion size could be well fitted by the inverse Weibull distribution and could be well applied to reveal the different inclusion size distribution based on the data examined by the rotated bending fatigue method. Finally, a new equation was proposed to establish the relationship among the loading stress amplitude, rotated bending fatigue number, and the maximum inclusion size.

**Keywords:** SAE52100 steel; melting route; maximum inclusion; rotated bending fatigue; Weibull distribution

## 1. Background

Bearing steel is an important material in mechanical structures, and its main products include rolling rods and rings of rolling bearings [1]. Bearing is a rotating part of equipment in machinery, railway, automobile, and other industries. The service life of bearing affects the service life of the entire equipment; therefore, the metallurgical quality and inspection standards of bearing steel have stringent requirements [2,3]. The cleanliness of bearing steel has a huge impact on its service life. The higher the cleanliness of the bearing steel, the smaller the size of the non-metallic inclusions in the bearing steel, and the higher the service life and reliability of the bearing steel. Therefore, the development trend of bearing steel is to produce cleaner steel with longer life and fewer inclusions [4,5].

The working environment of bearing steel is destined to be subjected to alternating loads in service; therefore, cracks easily appear near large particle inclusions and will be the source of crack initiation and expansion, continuously damaging the material [6,7]. Research shows that inclusions are one of the key factors causing the failure of bearing steel [8,9]. At present, the analysis and characterization of inclusions mainly include size, number, morphology, distribution, and chemical composition [10]. Non-metallic inclusions in steel can be detected by metallographic observation, electrolytic extraction, Aspex

automatic scanning electron microscope, and rotated bending fatigue test. The prepared sample for the metallographic observation method is first polished and then observed by placing it under a metallographic microscope. The size and morphology of the inclusions in the observed field of view are compared with the standard atlas provided by GB/T10561 (similar to the international standard ASTM E45:2013 and ISO 4967:2013), and each type of inclusion is rated. It has the advantages of low detection equipment requirements and fast detection speed but has high requirements for the surface cleanliness of polished samples. The observed surface showed random inclusion. The electrolytic extraction method exposes the inclusion after partial or complete dissolution of the sample, followed by analyzing the inclusion structure and composition. This method has the advantage of relatively complete observation of the appearance and size of the inclusion. In addition, the composition of the inclusion was analyzed by scanning electron microscopy (SEM) [11]. The disadvantage is that the dissolution takes a long time. Aspex adopts the integrated design of electron microscope and energy spectrum, controls the electron microscope and energy spectrum to realize automatic scanning of inclusions and synchronous analysis of chemical components [12]. The sample preparation and the observation surface of Aspex are the same as that of the metallographic observation method and thus also have the same disadvantages as that of the metallographic observation method. The advantages of Aspex include fast detection speed and relatively comprehensive detection results, including the location, type, size, composition, and morphology of inclusions [13].

In this study, extra clean bearing steels of SAE52100 obtained by the basic oxygen furnace plus ladle furnace plus vacuum degassing process (BOF + LF + RH) and vacuum induction melting (VIM) plus electroslag remelting (ESR) smelting methods were used as the test materials. Non-metallic inclusions were investigated by the metallographic observation method and Aspex automatic scanning method, and the results were verified by the rotated bending fatigue test. The results show that the metallographic and Aspex results are very similar, and the number of inclusions in the electroslag steel is more than that in the degassed steel in the same detection view, but the fatigue life of the degassed steel in the rotated bending test is actually lower than that in the electroslag steel. In order to address this problem, this study carried out in-depth research to further clarify the effect of different smelting processes on the type, size, and distribution of inclusions, and the influence of different types, sizes, and distribution of inclusions on bearing life.

## 2. Experimental

Two kinds of SAE52100 fully quenched bearing steels applied in this study were prepared by different smelting processes. The chemical compositions of the LF+RH steel and ESR steel are given in Table 1. The total oxygen content and the titanium content indicate a very high cleanliness. The heat treatment process of SAE52100 bearing steel is as follows: hot-rolled steel rods with a diameter of 60 mm were spheroidize annealed at 790 °C for 4 h, and cooled down to 720 °C for 2 h, then at 650 °C for 2 h, and finally air-cooled to room temperature. Samples were cut with a wire cutting machine, with the processing allowance reserved. Six samples were made for each of the two smelting processes of the two SAE52100 bearing steels, and the surface of every sample was grinned and polished using abrasive sand papers until the final mesh of 1000 to minimize the surface roughness. The metallographic method and Aspex explorer were carried out after polishing the ball bearing surface using a polishing machine. After each experiment, the samples were polished again (the thickness of the removed sample is not less than 0.1 mm) and repeated four times to obtain 24 sets of data to ensure the reliability of the prediction result of the Weibull distribution function. The maximum inclusion size was measured by both metallographic experiments and the Aspex explorer to quantitatively characterize the inclusions type, size, number, and distribution of the samples.

**Table 1.** The chemical compositions of two kinds bearing steels (wt.%).

| Sample | C | Si | Mn | P | S | Cr | Ni | Cu | Mo |
|---|---|---|---|---|---|---|---|---|---|
| LF + RH | 1.05 | 0.29 | 0.31 | 0.014 | <0.005 | 1.42 | 0.014 | 0.058 | <0.010 |
| ESR | 1.02 | 0.25 | 0.35 | 0.009 | <0.005 | 1.50 | 0.027 | 0.042 | 0.020 |

| Sample | Ti | Al | N | O | As | Ca | Pb | Sb | Sn |
|---|---|---|---|---|---|---|---|---|---|
| LF + RH | 0.0012 | 0.025 | 0.0019 | 0.0004 | <0.0050 | 0.0006 | 0.0001 | 0.0004 | 0.0004 |
| ESR | 0.0014 | 0.018 | 0.0053 | 0.0010 | <0.0077 | <0.0050 | 0.0001 | 0.0012 | 0.00017 |

The rotating bending fatigue test was carried out at room temperature in the mechanical fatigue test machine PQ1−6 (QianbangTest Equipment Co.LTD, Changchun, China) with a stress ratio R = −1 using a mechanical fatigue testing system with a resonance frequency of 80 Hz. Both rotated fatigue strengths limit at $1 \times 10^7$ cycles, and the S–N curve was obtained from the number of cycles and fracture conditions of 40 samples under corresponding loads, sample size was shown in Figure 1. After the specimen was fractured, the sample was rinsed with water for 30 s to remove stains during cutting. After ultrasonic cleaning in acetone solution for 20 min, the fracture surface was observed under SEM(FEI, Quanta 650FEG, OR, America) and the size and distribution of crack source of inclusion were measured. At the same time, the qualitative and quantitative analyses of the inclusion components were performed by Energy Disperse Spectroscopy (EDS Oxford, America).

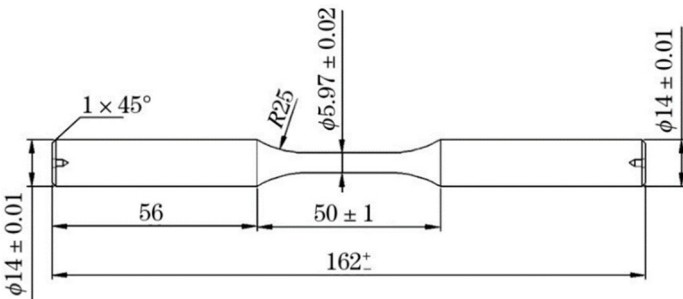

**Figure 1.** Illustration of sample dimension of the rotated bending fatigue test in this study.

### 3. Experimental Results

#### 3.1. Inclusions Evaluated by Metallography and Aspex Explorer

According to the standard of GB/T10561 (ASTM E45:2013 and ISO 4967:2013), the inclusion was qualitatively examined by the metallographic method according to the size, number, and category of the inclusions. Metallographic evaluation was performed on 24 groups of SAE52100 samples, and the worst results are listed in Table 2, indicating that the inclusions are divided into five categories as A (sulfide), B (aluminum oxide), C (metasilicate), D (oxide), DS (individual oxide), and the size level of the inclusion are divided into two levels of thin and thick. No clear difference in the inclusion evaluation could be identified by the metallographic method. The information of the size and the amount cannot be obtained by this semi-quantitative metallographic method, and thus, distinguishing the inclusion level of these two types bearing steel was difficult. In addition, as shown in Table 2, TiN was found in these two bearing steels and evaluated as inclusion by the categories of B, D, and DS, but did not show any clear difference.

**Table 2.** Metallographic observation rating results (taking the worst result as an example).

| Sample | Level | A | B | C | D | DS | B (TiN) | D (TiN) | DS (TiN) |
|---|---|---|---|---|---|---|---|---|---|
| LF + RH | thin | 0.5 | 0.5 | 0 | 0.5 | 1.0 | 0.5 | 0.5 | 1.0 |
| | thick | 0 | 0 | 0 | 0 | | 0 | 0.5 | |
| VIM + ESR | Thin | 0 | 0.5 | 0 | 0.5 | 1.0 | 0.5 | 0.5 | 1.0 |
| | thick | 0 | 0 | 0 | 0 | | 0 | 0.5 | |

The inclusions in the bearing steel of both LF + RH and VIM + ESR are revealed by type, size, number, and distribution scanned in a certain area by Aspex explorer. The results are listed in Figure 2, indicating that the number of the inclusion larger than 20 μm is higher in LF + RH steel than in VIM + VAR, whereas those with size smaller than 10 um size is lower in LF + RH than in VIM + VAR. Comparing with the metallographic method, the Asepx explorer is very useful for revealing the difference in the inclusions in terms of their size and distribution.

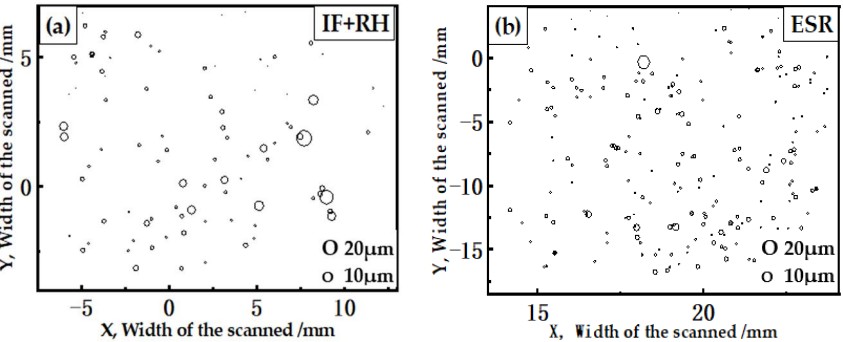

**Figure 2.** Distribution diagram of different types and sizes of inclusions in the two samples (**a**) LF + RH and (**b**) VIM + ESR.

In order to reveal the difference of inclusions in the two different bearing steels, the inclusions in both LF + RH and VIM + VAR were examined by finding the maximum inclusion in each examined field of the 24 samples. Figure 3a shows the typical maximum inclusion found in one of the 24 samples by the metallographic method, and Figure 3b shows a typical TiN inclusion found in one of the 24 samples examined by the Aspex explorer. The maximum inclusions examined by the metallographic method in the 24 samples are arranged in an ascending sequence, as shown in Figure 3c, and those examined by the Aspex explorer in the 24 samples are shown in Figure 3d. Figure 3a,b show that the non-metallic inclusions of LF + RH steel are mainly oxide inclusions, while that of ESR steel is mainly oxide inclusions and titanium nitride inclusions, both of which belong to brittle inclusions and are extremely unfavorable to fatigue life. Apart from the difference of the inclusion types, Figure 3c,d also show the size, indicating that irrespective of the method applied, the size of the bearing steel fabricated by LF + RH is clearly larger than that of the bearing steel fabricated by VIM + VAR.

### 3.2. Rotated Bending Fatigue Experiment Results

Figure 4a shows the S–N curves of LF + RH steel and ESR steel. It can be known from the same stress range (for the Y-axis coordinate), the data range of degassed steel is lower to the left, belonging to the low-stress range. The upper right side of the electroslag steel belongs to the high-stress range, indicating longer life of the electroslag steel under the same stress condition. The size range of inclusions observed at the fracture of electroslag steel during the rotary bending experiment is much smaller than that of the degassed steel, and that the rotated bending fatigue life of ESR steel is better than that of LF + RH steel at the same stress value. Among the 40 specimens of the LF + RH steel, a total of 34 samples were fractured, of which 31 cracks originated from the composite oxide inclusions, 1 is titanium nitride and 2 are other types of inclusions. Among the 40 samples of the ESR steel, a total of 20 were fractured. Among them, 10 cracks were induced by the composite oxide inclusions, 9 are titanium nitrides and 1 is another category. The rotated bending fatigue strength ($\sigma$-1) of LF + RH steel was calculated as 1000 MPa, and that of ESR steel is 1087 MPa.

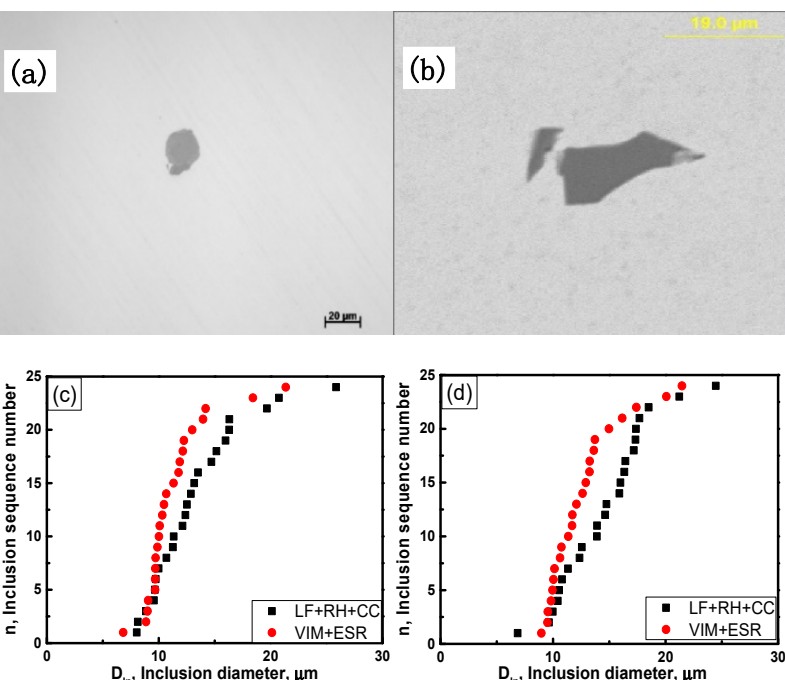

**Figure 3.** Maximum inclusions examined by both the metallographic method and Aspex explorer in the bearing steel of LF + RH and VIM + VAR. (**a**) Typical oxide inclusions (CaO-MgO-Al$_2$O$_3$) examined by the metallographic method, (**b**) typical TiN inclusions characterized by the Aspex explorer, (**c**) the maximum inclusions arranged in a descending sequence of the inclusion size for both LF + RH and VIM + ESR and (**d**) the maximum inclusions arranged in a descending sequence of the inclusion size for both LF + RH and VIM + ESR.

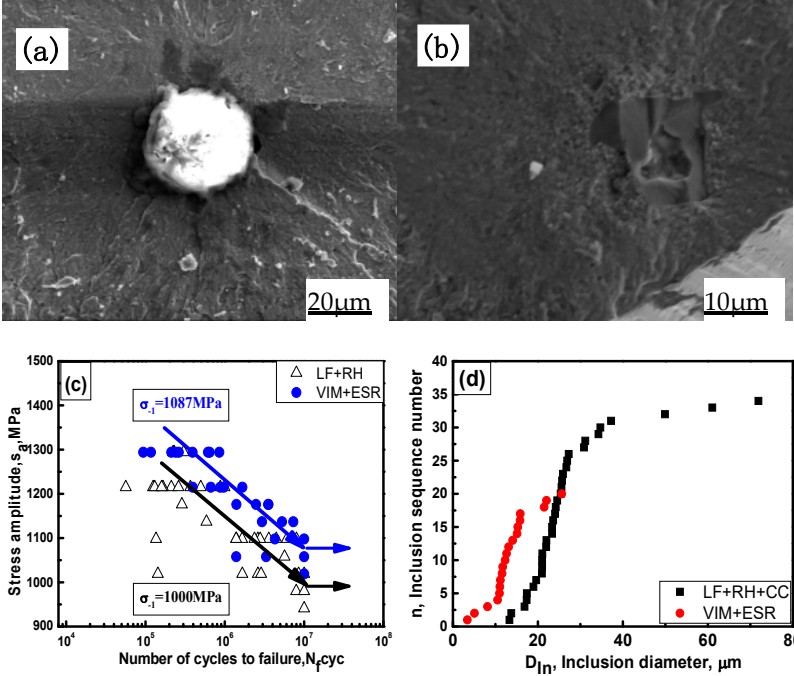

**Figure 4.** Rotated bending fatigue results. (**a**) Oxide inclusion in the fracture surface of the specimen with cycles of 1 under loading stress 1215 MPa, steel; (**b**) titanium nitride inclusion in the fracture surface of the specimen with cycles under loading stress 1098 MPa, (**c**) S–N curves of LF + RH steel and ESR, and (**d**) inclusion area distribution map obtained by Aspex detection.

The rotated bending fatigue test shows that the cracks in LF + RH steel and ESR steel generated by brittle non-metallic inclusions with large particles. Therefore, the brittle inclusions, which are harmful to the bearing steel, were selected, and the largest brittle inclusion was selected from each sample. The size of the largest inclusions in the obtained metallographic photo was measured by related software and shown in order from small to large. The distribution of inclusion size of $D_{In}$ is shown in Figure 4b in a size ascending sequence with $D_{In} = \sqrt{area}$.

## 4. Discussion

### 4.1. Accuracy Evaluation of Inclusion Distribution by Different Characterization Methods

The metallographic method has the advantage of quick detection of inclusions in bearing steel and, therefore, is a common technique for inclusion examination in the steel industry to evaluate the cleanliness of the special steel with inclusion categories A, B, C, D, and DS with thin and thick size levels [14,15]. It can quickly detect the type and size of non-metallic inclusions, but it is not very convenient to reveal and analyze the location distribution of inclusions, as shown in Table 2.

Aspex is an automatic inclusion detection device [16], which can quickly provide the type, size, composition, and position distribution of each inclusion in the bearing steel, as revealed in Figure 2, and thus it is a promising method to replace the metallographic method for the inclusion evaluation both in academia and the steel industry. The similarity of the inclusion distribution was obtained, and the results are shown in Figure 3 with the maximum inclusion size of 25 μm by both methods. In addition, the size of the inclusions in LF + RH is slightly larger than that in VIM + VAR, as shown in Figure 3c,d. The maximum inclusion size of the bearing steel produced by LF + RH is about 75 μm, as shown in Figure 4b and is about three times larger than that examined by both the metallographic method and Aspex explorer. However, both the distribution and size of the inclusions characterized by the rotated bending method are very close to that examined by the metallographic method and Aspex explorer, as shown in Figure 3a,b and Figure 4b.

In order to reveal the similarity and difference of the inclusion measurements by different methods, the distributions of the inclusions are replotted in Figure 5a for LF + RH and Figure 5b for VIM + ESR in the form of a Weibull distribution [17,18]. Figure 5a clearly shows a large difference for LF + RH, indicating that the inclusion size measured by the rotated bending fatigue method is about two times larger than that measured by the metallographic method and Aspex explorer. Figure 5b also shows very similar results for distribution and size for VIM + ESR. The big difference in the size distribution of LF + RH may be related to the different inclusion size measurement strategies, and the inclusion size measured by the rotated bending method has the biggest diameter of the inclusion initiated the crack [19]. The biggest inclusion size is seldom obtained in both the metallographic method and Aspex explorer owing to their two-dimensional sampling strategies [20,21]. However, this difference will be eliminated with increasing inclusion density, because it increases the chance to measure the biggest diameter of the inclusion due to the increased number of the inclusion in the examined section of the samples. Thus, the inclusion size distributions found by the three inclusion measuring methods are almost the same, as shown in Figure 5b. It should be pointed out here that the accuracy of the inclusion measurement is strongly affected by the examination method, and thus the rotated bending method is one of the very reliable methods for inclusion characterization because it is derived from the fracture surface of the rotated bending fatigue specimens.

Apart from the accuracy of the measurement of the inclusion, the assumption of the Weibull function cannot be applied to reveal the distribution of the maximum inclusion size because of the significant deviation from the straight line, as shown in Figure 5a,b. However, when the proposed inverse Weibull distribution is applied to describe the relationship between the Weibull strength and the inverse value of $D_{In}$ [22,23], a very beautiful linear relationship could be obtained, as shown in Figure 5c,d. Thus, it could be concluded that the maximum inclusion size could be accurately characterized by the rotated bending

fatigue test, and the distribution of the maximum inclusion size could be well described by the inverse Weibull distribution, as proposed in our previous research [24].

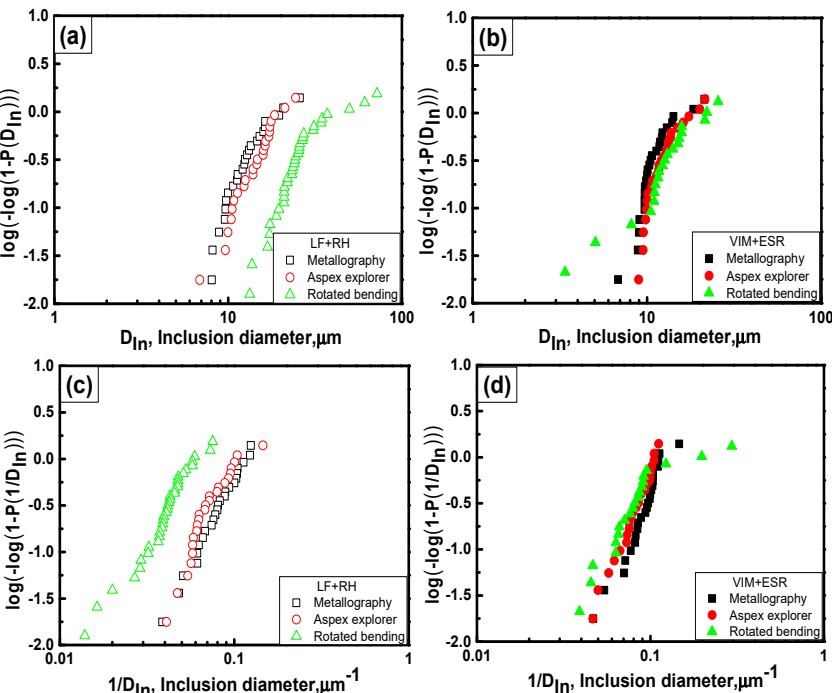

**Figure 5.** Inclusion Weibull distribution and inverse Weibull distribution of bearing steel of both LF + RH and VIM + ESR evaluated by the Metallographic method, Aspex explorer, and rotated bending fatigue. (**a**) LF + RH revealed by the Weibull distribution, (**b**) VIM + ESR revealed by the Weibull distribution, (**c**) LF + RH revealed by the inverse Weibull distribution and (**d**) VIM + ESR revealed by the inverse Weibull distribution.

### 4.2. Relationship among Fatigue Stress Amplitude, Inclusion Size and Fatigue Cycles

Figure 4a shows the relationship between the stress amplitude and the rotated bending fatigue cycle number, in which only the fatigue cycle effect is considered. It is well known that the rotated bending strength is not only the function of fatigue cycle number but also the internal factor of inclusion as well. Detailed analysis of the effects of both cycle number and inclusion size was carried out. The relationship between the stress amplitude ($\sigma_a$) and the fatigue cycle number (*n*) could be well described by the Basquin law, as shown in Figure 4a [25,26], indicating that the stress amplitude at given fatigue cycles is much higher for VIM + ESR than that of LF + RH. This higher fatigue strength could be related to the inclusion size, as revealed in Figure 5a,c. In order to build the relationship among the stress amplitude, the inclusion size, and the fatigue cycles, the fatigue failure life (*N*) is plotted as a function of the inclusion size ($D_{In}$), as shown in Figure 6a, by choosing the data of *N* and $D_{In}$ under a given loading stress of 1098, 1215, and 1294 MPa. Under a given loading stress amplitude, the rotated bending fatigue life decreases significantly with increasing inclusion size. The relationship between the fatigue life as a function of the inclusion size is calculated based on Equation (1).

$$N = k(\sigma_a)D_{In}^{-\alpha} \tag{1}$$

where *N* is the fatigue cycle number at a given loading stress of $\sigma_a$, and $D_{In}$ is the inclusion size; $k(\sigma_a)$ is a function of loading stress amplitude and $D_{In}$ is the inclusion size measured by the rotated bending test; $\alpha$ is the exponent of inclusion.

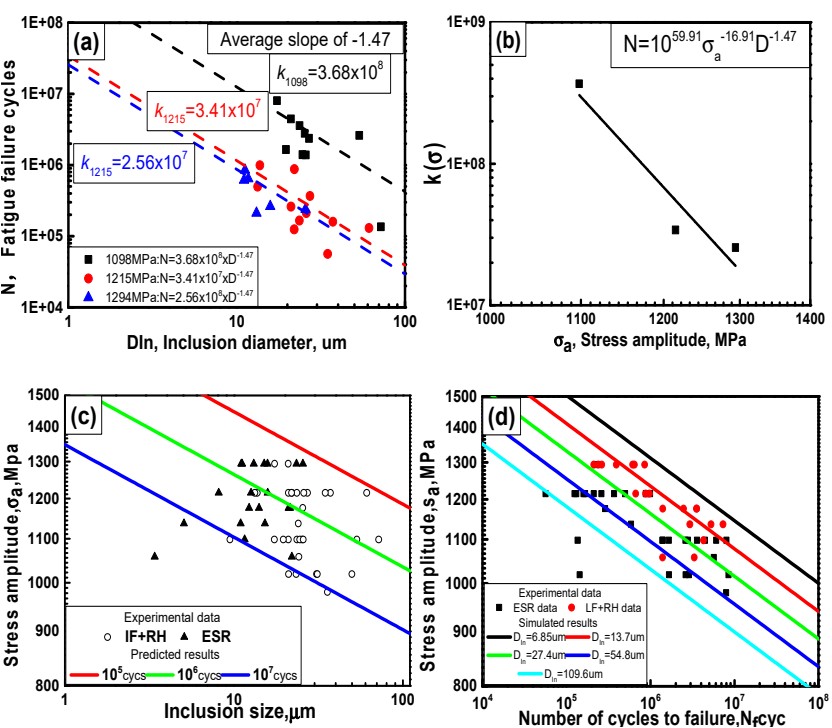

**Figure 6.** Experimental and simulated results on the relationship between the stress amplitude, fatigue cycle number, and inclusion size. (**a**) Experimental data and calculation of the dependence of fatigue cycle number on inclusion size measured under different loading stress amplitudes, (**b**) calculated relationship between the fatigue cycle number, stress amplitude, and inclusion size based on data from (**a**), (**c**) comparison of the experimental data and the predicted results by Equation (3) regarding to the dependence of loading stress amplitude and inclusion size and (**d**) comparison of the experimental data and the predicted results by Equation (3) regarding the dependence of loading stress amplitude and fatigue cycle number.

The exponent of σ is calculated based on the fitting of Equation (1) and the average value of −1.47 was applied to calculate the $k\,(\sigma_a)$, as shown in Figure 6a,b. The $k\,(\sigma_a)$ is shown as a function of loading stress amplitude of $\sigma_a$, as revealed in Figure 6b. The relationship between the stress amplitude, fatigue cycle number, and inclusion size could be obtained as described by Equation (2).

$$N = C_1 \sigma_a^{-\beta} D_{In}^{-\alpha} \tag{2}$$

where $N$ is the fatigue cycle number at a given loading stress of $\sigma_a$ and $D_{In}$ is the inclusion size; $D_{In}$ is the inclusion size measured by the rotated bending test; $\alpha$ is the exponent of inclusion, $C_1$ is a constant, $\beta$ is the exponent of loading stress amplitude.

Based on the calculation, $C_1 = 8.13 \times 10^{59}$, $\beta = -16.91$ and $\alpha = -1.47$ were derived from the experimental data under different loading stress amplitudes of 1215, 1098, and 1294 MPa. Equation (2) could be rewritten in the form of Equation (3) with $\delta = 1/\beta = 0.059$, $\gamma = \alpha/\beta = 0.087$, and $C_2 = 3490$. The constants are summarized in Table 3.

**Table 3.** Constants summary.

| Constants | $\alpha$ | $C_1$ | $\beta$ | $C_2$ | $\delta$ | $\gamma$ |
|---|---|---|---|---|---|---|
| | −1.47 | $8.13 \times 10^{59}$ | −16.91 | 3490 | 0.059 | 0.087 |

The comparison between the experimental results and the simulated results based on Equation (3) is revealed in Figure 6c,d, proving the accuracy of Equation (3) for the

prediction of the relationship between the stress amplitude, the inclusion size, and the fatigue cycles.

$$\sigma_a = C_2 N^{-\delta} D_{In}^{-\gamma} \tag{3}$$

where $N$ is the fatigue cycle number at a given loading stress of $\sigma_a$, and $D_{In}$ is the inclusion size; $D_{In}$ is the inclusion size measured by the rotated bending test; $\alpha$ is the exponent of inclusion, $C_2$ is a constant, $\gamma$ is the exponent of the inclusion, and $\delta$ is the exponent of the rotated fatigue cycle number.

## 5. Conclusions

Based on the metallographic observation experiment, Aspex inclusion detection experiment, and rotated bending fatigue test, the size of the largest non-metallic inclusions in bearing steel was examined, and the correspondence relationship between the rotated bending fatigue life ($N$), non-metallic inclusions size ($D_{In}$), and the loading stress amplitude ($\sigma_a$) is proposed. The main conclusions of this study are summarized as follows.

(1) The non-metallic inclusions in both LF + RH steel and ESR steel were characterized by the metallographic method, Aspex explore, and rotated bending fatigue test, indicating that both the metallographic method and Aspex explorer underestimate the size of the maximum inclusions. However, the rotated bending fatigue method successfully examined the maximum inclusion size.
(2) The distribution of the maximum inclusions could not be described by the classical Weibull distribution based on the inclusion size, whereas the inverse Weibull distribution of the maximum inclusion size could be well applied based on the inverse value of the maximum inclusions.
(3) The rotated bending fatigue life is not only determined by the loading stress amplitude but also by the maximum inclusion size. The relationship between the rotated bending fatigue cycle number, the loading stress amplitude, and the maximum inclusion size was established and shown to accurately predict the dependence among these three parameters.

**Author Contributions:** Conceptualization, H.X., C.W. and W.C.; Data curation, X.A., Z.S., H.X. and Y.W.; Formal analysis, X.A., Z.S., C.W., W.C. and J.Y.; Funding acquisition, Y.W.; Investigation, X.A., Z.S. and H.X.; Methodology, X.A., Z.S., H.X., C.W., Y.W. and J.Y.; Project administration, H.X., C.W., Y.W., W.C. and J.Y.; Resources, W.C.; Supervision, W.C.; Validation, X.A. and J.Y.; Writing—original draft, X.A.; Writing—review & editing, X.A. All authors have read and agreed to the published version of the manuscript.

**Funding:** This research is supported by the Central Iron and Steel Research Institute independently invested in a special research and development fund (No. shi20T61200ZD), National Natural Science Foundation of China (NSFC) (Nos. 51871062 and 51871194).

**Institutional Review Board Statement:** Not applicable.

**Informed Consent Statement:** Not applicable.

**Data Availability Statement:** The data presented in this study are available on request from the corresponding author.

**Acknowledgments:** The authors want to thank State Key Laboratory of Metastable Materials Science and Technology, Yanshan University, Qinhuangdao 066004, Hebei, & Steel Research Institute for the supply of material and technical support in this work.

**Conflicts of Interest:** The authors declare no conflict of interest.

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
