# Peer review of "Quantitative Examination of the Inclusion and the Rotated Bending Fatigue Behavior of SAE52100"

_metals, doi:10.3390/met11101502_

Round 1
Reviewer 1 Report
The results are interesting. The present submission reports the effect of inclusion on properties and the rotated bending fatigue behavior of SAE2100. By careful metallographic observation, the investigators were able to find the max size of non-metallic inclusions which may show a correlation with properties from the bending fatigue experiments. It is a good piece of work.
There are a few minor comments.
Minor comments
-Please add more details on the experimental. Especially, on the rotating bending fatigue test.
-Please use the full name in the first instance, even in the abstract.
ex) VIM to vacuum induction melting (VIM)
-Please double-check the number of Figures.
ex) line 152, “Figure 3a shows the SN curves”
Author Response
Dear Editor,
First of all, we would like to thank you and the reviewers of our paper (Manuscript Number: Metals-1376694) for the constructive and informative comments and suggestions. We have carefully considered the reviewers’ comments and suggestions and have modified our manuscript accordingly. The following is our detailed response to the reviewers’ comments and suggestions. For clarity, we have listed the reviewers’ comments below and have addressed them one by one.
Response to Reviewer 1 Comments
Point 1: Please add more details on the experimental. Especially, on the rotating bending fatigue test.
Response 1: Thank for your constructive suggestion. Considering the reviewer’s suggestion, we have added two inclusion pictures of rotary bending fatigue experiments, please see for Line 188 Figure 4. (a) and (b).
Point 2: Please use the full name in the first instance, even in the abstract. ex) VIM to vacuum induction melting (VIM)
Response 2: Thank for your carefully work. We complete all abbreviations.
Point 3: Please double-check the number of Figures. ex) line 152, “Figure 3a shows the SN curves”
Response 2: Thank for your carefully work. The error has been corrected.
Finally, thank you for giving us a chance to revise our manuscript. Certainly, we also thank the review’s comments and suggestions. At last, we hope this revised manuscript could bring more to reader.
Yours sincerely,
Xueliang An,
Author
State Key Laboratory of Metastable Materials Science and Technology, Yanshan University, Qinhuangdao 066004, Hebei, P. R. China
E-mail: anxueliang@stumail.ysu.edu.cn; caowenquan@nercast.com

Reviewer 2 Report
The manuscript under review elaborates a quantitative examination of the inclusion and their correlation to the rotated bending fatigue behavior of SAE52100. Useful information are given to the reader however there are several sections of the paper that need improvements before accepting it for publication. More specifically: Abbreviations ASPEX, LF+RH, BOF+LF+RH must be explained once in the text Page 3, Table 1: It seems that the 5th and 6th rows are identical to the 2nd and 3rd rows. Page 3, line 99:"were measured" Please use an alternative expression for the construction of the S-N curve Page 3, line 100:"the fracture surface was cleaned" Please explain the method employed for the cleanning of the surface Page 4, line 106: "According to the standard of GB/T10561 of China" It is preferable the authors provide an International standard and its relevant reference. Page 4, line 109: "and the results are listed in Table 2" How many measurements were performed by the authors? At which magnification? Page 4, Figure 2: What is ABS/mm written in the axes title? Page 6, line 160-163: "Among 40 specimens...other category" The authors are advised to provide Figures (either from SEM or optical microscopy to support the statement. My overall recommendation is major revision
Author Response
Dear Editor,
First of all, we would like to thank you and the reviewers of our paper (Manuscript Number: Metals-1376694) for the constructive and informative comments and suggestions. We have carefully considered the reviewers’ comments and suggestions and have modified our manuscript accordingly. The following is our detailed response to the reviewers’ comments and suggestions. For clarity, we have listed the reviewers’ comments below and have addressed them one by one.
Response to Reviewer 2 Comments
Point 1: Abbreviations ASPEX, LF+RH, BOF+LF+RH must be explained once in the text Page 3,
Response 1: Thank for your carefully work. We complete all abbreviations.
Point 2: Table 1: It seems that the 5th and 6th rows are identical to the 2nd and 3rd rows.
Response 2: Thank for your carefully work. The error has been corrected, please see for Lines 97-98.
Point 3: Page 3, line 99:"were measured" Please use an alternative expression for the construction of the S-N curve Page 3,
Response 3: Thank for your carefully work. The error has been corrected, please see for Lines 104-105.
Point 4: line 100:"the fracture surface was cleaned" Please explain the method employed for the cleanning of the surface Page 4,
Response 4: Thank for your constructive suggestion. The cleaning method has been written down, please see for Lines 106-109.
Point 5: line 106: "According to the standard of GB/T10561 of China" It is preferable the authors provide an International standard and its relevant reference.
Response 5: Thank for your constructive suggestion. The international standards have been added, please see for Lines 48 and 115-116.
Point 6: Page 4, line 109: "and the results are listed in Table 2" How many measurements were performed by the authors? At which magnification?
Response 5: Thank for your constructive suggestion. The statement in Line 118 in the article may have been inaccurate enough to be misleading, and we have corrected it.
We did measure 24 sets of data, please see for Lines 90-93 “After each experiment, the samples were polished again (the thickness of the removed sample is not less than 0.1 mm), And repeating four times to obtain 24 sets of data to ensure the reliability of the prediction result of the Weibull distribution function.”
Point 7: Page 4, Figure 2: What is ABS/mm written in the axes title? Page 6,
Response 3: Thank for your carefully work. The “ABS/mm” in Figure 2 (Line 135) was the distance that Aspex scans. We were changed the “ABS/mm” to scan distances.
Point 8: line 160-163: "Among 40 specimens... other category" The authors are advised to provide Figures (either from SEM or optical microscopy to support the statement.
Response 8: Thank for your constructive suggestion. Considering the reviewer’s suggestion, we have added two inclusion pictures of rotary bending fatigue experiments, please see for Line 188 Figure 4. (a) and (b).
Finally, thank you for giving us a chance to revise our manuscript. Certainly, we also thank the review’s comments and suggestions. At last, we hope this revised manuscript could bring more to reader.
Yours sincerely,
Xueliang An,
Author
State Key Laboratory of Metastable Materials Science and Technology, Yanshan University, Qinhuangdao 066004, Hebei, P. R. China
E-mail: anxueliang@stumail.ysu.edu.cn; caowenquan@nercast.com

Reviewer 3 Report
Dear Authors,
Thank you for sending your paper for publication. In your paper, you compare three methods for inclusion detection, and by the way, you present the effect of the maximum inclusion on the fatigue life of selected bearing steel. The result of the inclusion influence was easy to predict but I like the comparison of the methods. During reading, I found some problems that need to be corrected.
* You missed two references (2 and 14).
Lines 75-78: This singular sentence should be divided into three: first ending after the word "processes" and second ending after "Table 1".
Line 77: should be "are" instead of "is".
Line 81: should be "cut with" instead "cut".
Line 92: Table 1 is repeated.
Line 96: should be "rotating" instead "rotated".
Line 97: "of" is not necessary.
Line 99: you missed a superscript in the number of cycles.
Line 103: unexplained abbreviation EDS.
Line 118: For me, Table 2 is not enough explained, I mean what numbers in the table mean.
Line 137: you wrote that Figure 3 shows that non-metallic inclusions are mainly oxide inclusions, well, in this figure I can see only examples so the word "mainly" does not fit.
Line 152: should be "Figure 4a".
Line 153: remove "in".
Line 173: as you always start figure captions with capital letters there should be "Carbon".
Line 188: should be "results" instead of "resulted".
Lines 261-263: you put all equations in one place; I prefer to locate them right after the text that refers to them, I know they in fact would be close to each other anyway but now you repeated the explanation for inclusion size Din.
You present constants for your equations in the text. They would be more readable if you put them into a Table.
Best regards,
Author Response
Dear Editor,
First of all, we would like to thank you and the reviewers of our paper (Manuscript Number: Metals-1376694) for the constructive and informative comments and suggestions. We have carefully considered the reviewers’ comments and suggestions and have modified our manuscript accordingly. The following is our detailed response to the reviewers’ comments and suggestions. For clarity, we have listed the reviewers’ comments below and have addressed them one by one.
Response to Reviewer 3 Comments
Point 1: * You missed two references (2 and 14).
Response 1: Thank for your carefully work. The error has been corrected.
Point 2: Lines 75-78: This singular sentence should be divided into three: first ending after the word "processes" and second ending after "Table 1".
Response 2: Thank for your carefully work. The error has been corrected, please see for Lines 79-82.
Point 3: Line 77: should be "are" instead of "is".
Response 3: Thank for your carefully work. The error has been corrected, please see for Line 79.
Point 4: Line 81: should be "cut with" instead "cut".
Response 4: Thank for your carefully work. The error has been corrected, please see for Line 85.
Point 5: Line 92: Table 1 is repeated.
Response 5: Thank for your carefully work. The error has been corrected, please see for Table 1 in the Line 98.
Point 6: Line 96: should be "rotating" instead "rotated".
Response 7: Thank for your carefully work. The error has been corrected, please see for Line 101.
Point 8: Line 97: "of" is not necessary.
Response 8: Thank for your carefully work. The error has been corrected, please see for Line 102.
Point 9: Line 99: you missed a superscript in the number of cycles.
Response 9: Thank for your carefully work. The error has been corrected, please see for Line 104.
Point 10: Line 103: unexplained abbreviation EDS.
Response 10: Thank for your carefully work. We complete all abbreviations.
Point 11: Line 118: For me, Table 2 is not enough explained, I mean what numbers in the table mean.
Response 11: Thank for your constructive suggestion. The statement in Line 118 in the article may have been inaccurate enough to be misleading, and we have corrected it.
We did measure 24 sets of data, please see for Lines 90-93 “After each experiment, the samples were polished again (the thickness of the removed sample is not less than 0.1 mm), And repeating four times to obtain 24 sets of data to ensure the reliability of the prediction result of the Weibull distribution function.”
Point 12: Line 137: you wrote that Figure 3 shows that non-metallic inclusions are mainly oxide inclusions, well, in this figure I can see only examples so the word "mainly" does not fit.
Response 12: Thank for your carefully work. The "mainly" in Figure 3 (Line 156 and Line 157) may have been inaccurate enough to be misleading, and we have corrected it., please see for Lines 156-157.
Point 13: Line 152: should be "Figure 4a".
Response 13: Thank for your carefully work. The error has been corrected, please see for Line 162.
Point 14: Line 153: remove "in".
Response 14: Thank for your carefully work. The error has been corrected, please see for Line 163.
Point 15: Line 173: as you always start figure captions with capital letters there should be "Carbon".
Response 15: Thank for your carefully work. The error has been corrected, please see for Lines 185-189.
Point 16: Line 188: should be "results" instead of "resulted".
Response 16: Thank for your carefully work. The error has been corrected, please see for Line 202.
Finally, thank you for giving us a chance to revise our manuscript. Certainly, we also thank the review’s comments and suggestions. At last, we hope this revised manuscript could bring more to reader.
Yours sincerely,
Xueliang An,
Author
State Key Laboratory of Metastable Materials Science and Technology, Yanshan University, Qinhuangdao 066004, Hebei, P. R. China
E-mail: anxueliang@stumail.ysu.edu.cn; caowenquan@nercast.com

Round 2
Reviewer 2 Report
The manuscript has been considerably improved following the modifications made by the authors.